# Breakup Processes and Droplet Characteristics of Liquid Jets Injected into Low-Speed Air Crossflow

**Lingzhen Kong [1,2,\*], Tian Lan [3], Jiaqing Chen [1,2], Kuisheng Wang [3] and Huan Sun [1]**

[1] School of Mechanical Engineering, Beijing Institute of Petrochemical Technology, Beijing 102617, China; jiaqing@bipt.edu.cn (J.C.); sunhuan0621@163.com (H.S.)

[2] Beijing Key Laboratory of Pipeline Critical Technology and Equipment for Deep Water Oil & Gas Development, Beijing 102617, China

[3] College of Mechanical and Electrical Engineering, Beijing University of Chemical Technology, Beijing 100029, China; bruskylan@163.com (T.L.); wangks@mail.buct.edu.cn (K.W.)

\* Correspondence: konglingzhen@bipt.edu.cn

**Abstract:** The breakup processes and droplet characteristics of a liquid jet injected into a low-speed air crossflow in the finite space were experimentally investigated. The liquid jet breakup processes were recorded by high-speed photography, and phase-Doppler anemometry (PDA) was employed to measure the droplet sizes and droplet velocities. Through the instantaneous image observation, the liquid jet breakup mode could be divided into bump breakup, arcade breakup and bag breakup modes, and the experimental regime map of primary breakup processes was summarized. The transition boundaries between different breakup modes were found. The gas Weber number ($We_g$) could be considered as the most sensitive dimensionless parameter for the breakup mode. There was a $We_g$ transition point, and droplet size distribution was able to change from the oblique-I-type to the C-type with an increase in $We_g$. The liquid jet Weber number ($We_j$) had little effect on droplet size distribution, and droplet size was in the range of 50–150 μm. If $We_g > 7.55$, the atomization efficiency would be very considerable. Droplet velocity increased significantly with an increase in $We_g$ of the air crossflow, but the change in droplet velocity was not obvious with the increase in $We_j$. $We_g$ had a decisive effect on the droplet velocity distribution in the outlet section of test tube.

**Keywords:** liquid jet; primary breakup; breakup regime; PDA; droplet size; droplet velocity

## 1. Introduction

Atomizing the injected liquid through the air flow in a tube can significantly increase the gas–liquid contact surface area, and the atomized micron droplets are easier to disperse uniformly in the air flow, which can strengthen the gas–liquid contact mass transfer process. Liquid jet atomization mixing technology has been extensively applied in novel gas–liquid contact absorption equipment in the energy and chemical industry areas, such as the nozzle type Venturi scrubber in gas absorption purification process [1–3], the tubular gas–liquid atomization mixed contactor in the natural gas dehydration process [4,5] and so on. The favorable spray characteristic of a liquid absorbent is the premise to ensure excellent mass transfer efficiency of the tubular gas–liquid contact absorber, because the micron droplets in the airflow have good dispersive characteristics and a high surface area for absorption of the contaminant. The liquid solvent is injected into the air flow through small holes in the tube wall and then atomized by the gas–liquid interaction, which causes a low–pressure drop, but a highly efficient atomization. Such methods are commonly used in compact absorption devices. The deformation and atomization process of a liquid jet induced by low-speed air flow is one of the core issues in designing a high-performance tubular droplet generator. In order to optimize the structure and performance of

the droplet generator of a tubular absorber, it is essential to identify and understand the interacting behavior between the liquid jet and the cross airflow inside the process pipeline.

Liquid jet injected into a gas crossflow undergoes bending, deformation and fracture processes, which are known as liquid jet primary breakup. With the further interaction between gas and liquid, the large liquid blocks continue to break up into smaller droplets, demarking the secondary breakup of the liquid jet and ultimately forming the atomization structure [6]. The uniformity of the atomization structure is a critical prerequisite for the efficiency of gas–liquid mass transfer. Depending on the gas–liquid flow conditions, there could be several primary breakup mechanisms of liquid jets breaking into ligaments, bags, and droplets with different sizes and velocities.

The complicated mechanisms resulting from the interaction of a liquid jet with a gas crossflow are still an attractive research topic for many researchers and engineers internationally. Fuller et al. and Wu et al. [7,8] have analyzed the physical state of liquid jet injected into a subsonic air crossflow under different working conditions by a high-speed camera. They proposed the breakup modes map based on the liquid–gas momentum flux ratio (q) and gas Weber number ($We_g$). It is believed that there are two main breakup modes, namely, column breakup and surface breakup, and the borderline of these breakup modes is given. Following the experimental study of Wu and Tambe [9], Lee [10], and Bolszo [11] further demonstrate the common applicability of the breakup modes map based on q and $We_g$, and draw a conclusion that $We_g$ is the most effective dimensionless parameter for determining breakup modes. The other type of map, the $We_g$–$We_j$ regime map, is used to classify breakup characteristics of liquid jet injected into air crossflows, which is first proposed by Vich and Ledoux [12]. It is different from the breakup state of liquid injection under the action of a high–speed crossflow studied by Wu et al.; they believe that the modes of liquid jet breakup under the condition of low–speed transverse flow can be divided into breakup without transverse flow, arcade breakup and bag breakup. Birouk et al. [13] also observed a similar breakup mode in their experimental studies. Birouk gives a transition range of these two modes based on the $We_g$–$We_j$ regime map proposed by Vich and Ledoux [12]. Additionally, there are other various types of breakup regime maps in the literature, such as the Ohnesorge number (Oh)–$We_g$ [14,15] and $We_g$–$\lambda_s/d_j$ [16]. For all of these maps, $We_g$ is the main dimensionless parameter for defining the primary breakup modes under different transverse flow conditions.

Liquid jet penetration has a direct impact on the distribution of droplets in a tubular absorber, and accordingly its absorption and mixing rate with gas. This is also important for the design of a tubular absorber as to prevent impingement of atomized droplets on its walls. Liquid jet penetration has always been a popular subject in the research of transverse liquid jets. Wu et al. [7] deduced the theoretical formula of jet penetration trajectory, in which the penetration depth is 0.5 power of the power of the momentum flux ratio. Similar conclusions are also obtained in their later experimental study. Tambe [17], Brown [3] and Song [18] have reported results correlating to the trajectories of upper liquid jet boundaries via experimental research. Notably, the formulas fitted by different scholars are different in form and correlation coefficient [19,20]. In general, these differences are due to a number of factors, including trajectory estimation method, liquid properties (such as density, viscosity, surface tension), test conditions (such as temperature and pressure), internal geometry of the jet orifice (such as flow coefficient, cavitation behavior), measurement methods (pulse shadow, high–speed photography), and so on [21]. Thus, it is urgent and challenging to accurately formulate a liquid jet trajectory prediction formula by a rational method.

Estimations for both droplet size and velocity distributions are crucial to evaluate the atomization performance of the tubular gas–liquid mixer, as well as for feedback on the design and subsequent theoretical research. The methods used in this experiment are mostly noncontact optical measuring techniques, such as phase-Doppler anemometry (PDA), Malvern, and so on, which can avoid the influence of the measuring technique itself on the flow field. Because liquid injected atomization is influenced by various factors, the dimensionless parameters are introduced to evaluate the atomization performance, and the synergistic effect between diverse parameters on the atomization effect is adopted.

Hautman and Rosfjord [22] have used Malvern to study the atomization characteristics of a liquid jet injected into a subsonic air crossflow, which concluded that the increase in the air flow rate and air density, and the reduction in the liquid surface tension can be responsible for enhancing liquid jet atomization and reducing the size of the droplets in the flow field. Tambe et al. [9,17] assume that droplet size decreases when the air crossflow velocity increases, and the droplet size in the center of the liquid jet plume for low momentum ratios or near the periphery for high momentum ratios is the largest. Although considerable progress has been made regarding droplet size characteristics, some controversy and intriguing findings have emerged. Reichel et al. [23] think that the droplet size distribution is layered, and the droplet size is gradually increased with distance from the wall. Miller et al. [24] observed that the liquid–gas momentum flux ratio q has less effect on the droplet's Sauter mean diameter (SMD) through digital holographic diagnostic techniques. The effect of $We_g$ ($We_g$ = 33–2020) on the atomization properties is investigated by Lubarsky et al. [25]. They noticed that droplet diameters are found to be in the range of 15–30 μm for higher $We_g$, while larger droplets (100–200 μm) are observed at the lower values of $We_g$. The droplet diameter exhibits an S-shaped distribution, which means the droplet diameter is smaller inside the plume, and larger near the outer edge. Further fragmentation along the direction of the gas flow makes the S-shaped particle size distribution to gradually change to a C-shaped distribution and shows the potential to develop I-shaped distribution. In addition, the particle size is more uniform and smaller along the liquid jet direction. Gradually, some correlations of the mean droplet diameters have been summarized by the fitting relationship [18,26].

Almost all of the aforementioned studies have summed the breakup processes of liquid jets that are injected into the subsonic or a supersonic air crossflow from a nozzle, and the droplet sizes and velocities of the jets are also measured in these studies. However, gas velocity in industrial pipeline of energy and chemical industry is usually between 8 m/s and 40 m/s. The influence of aerodynamic force on the breakup of liquid jet is weak, and the liquid jet primary breakup mode under the action of a low-speed gaseous crossflow is clearly different from that of a subsonic or supersonic gaseous crossflow. Whether the breakup characteristics and droplet distribution of liquid jets, obtained under the condition of a subsonic or supersonic gaseous crossflow, are suitable for the condition of low–speed gas flow in industrial pipeline remains to be verified. The objective of this paper is to clarify the effects of the gas Weber number ($We_g$) and the liquid jet Weber number ($We_j$) on the breakup and atomization processes of liquid jets in a low-speed crossflow of air. The breakup processes of liquid jets are recorded by high-speed photography, and phase-Doppler anemometry (PDA) is employed to measure the droplet sizes and droplet velocities. The results could contribute to the structure design of a tubular gas–liquid atomization mixer and the selection of applicable numerical simulation models.

## 2. Materials and Methods

### 2.1. Experimental Materials

As shown in Figure 1, an experimental apparatus is specially built to study the atomization mechanism of liquid jets in an air crossflow. The experimental system consists of three parts: a gas supply system, a liquid injection system and an experimental measurement system. The experiments are performed at atmospheric pressure and room temperature, and the working fluids are water and air. The compressed air is provided by an air compressor which is connected with a high-pressure tank. The vortex gas flowmeter (VAFTP-050-DC1-213-1.0-P1.6) which has a temperature and pressure compensation function is used to measure the gas flow. The orifice type gas distributor, installed in the upstream of the experimental section, ensures that the airflow velocity for the inlet of the experimental section is uniformly distributed in the cross section of the test tube. During the experiment, the water is supplied from a pressurized vessel to the liquid nozzle. The pressure in the vessel is adjusted to the set value by adjusting the pressure of the reducing valve. The flow rate of the liquid is measured by an electromagnetic flow meter and controlled by a high-precision flow controller. High-speed

photography and PDA are separately used to visualize the breakup process of the liquid jet and simultaneously measure the size and velocity of droplets, respectively. The atomized droplets are collected by the water tank at the outlet of the test section.

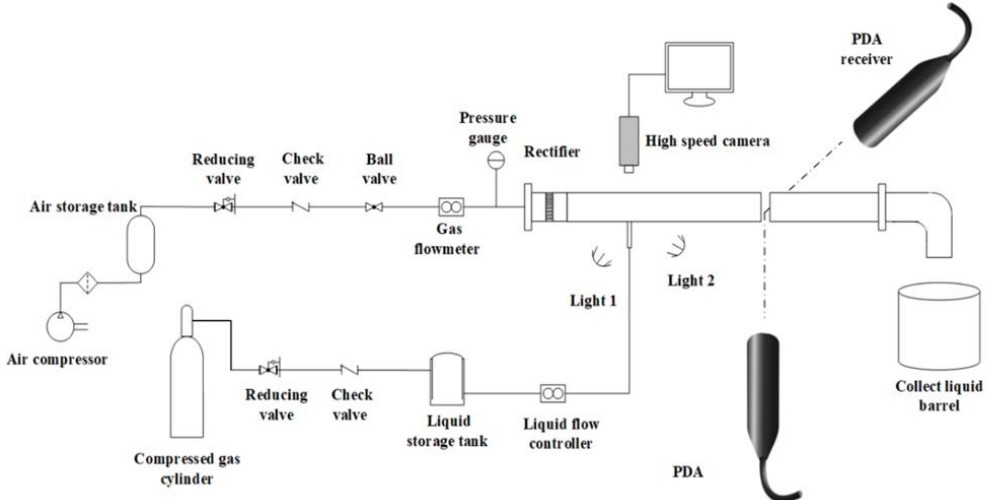

**Figure 1.** Schematic of experimental apparatus.

Figure 2 depicts details of the experimental measurement section and the employed coordinate system. The test tube has a rectangular cross section of 50 mm × 50 mm and a length of 800 mm. The test tube is made of clear acrylic resin that allows flow visualization and imaging of the events. The jet orifice is mounted on the center of the lower surface of the plexiglass square tube. The end wall surface of the jet orifice is flush with the inner surface of the tube wall. The details of the jet nozzle used in the experiment are delineated in Figure 3. The jet orifice has a diameter of $d_j = 1$ mm and a length to diameter ratio of $L/d_j = 4$. In the experiment, the reference coordinate system takes the jet nozzle center as the origin, where the X coordinate is in the direction of the air crossflow, the Y coordinate is in the direction of the width of the test tube, and the Z coordinate is in the direction of liquid jet injection.

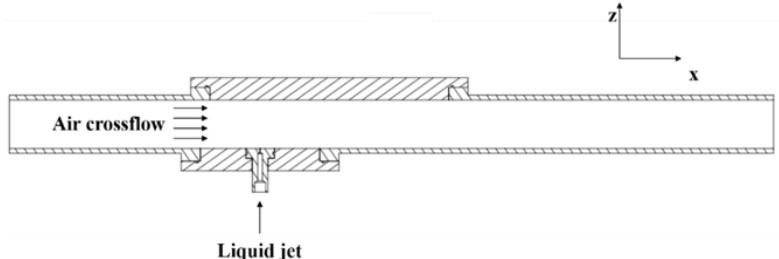

**Figure 2.** Schematic of the test section.

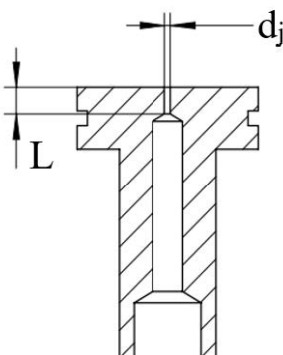

**Figure 3.** Details of the jet orifice.

The experimental conditions explored in the present investigation are summarized and shown in Table 1. The crossflow velocity is calculated by the formula $u_g = Q_g/A_g$, where $Q_g$ is the measured gas flow rate and $A_g$ is the cross-sectional area of the square tube. The liquid jet velocity is also calculated by the formula $u_j = Q_j/A_j$ where $Q_j$ is the measured liquid flow rate and $A_j$ is the cross-sectional area of the liquid jet orifice.

**Table 1.** Experimental conditions.

| Experiment Parameter | Number |
|---|---|
| Liquid jet velocity $u_j/m·s^{-1}$ | 0–10 |
| Liquid density $\rho_j/kg·m^{-3}$ | 997 |
| Air crossflow velocity $u_g/m·s^{-1}$ | 5–30 |
| Air density $\rho_g/kg·m^{-3}$ | 1.17 |
| Surface tension $\sigma/N·m^{-1}$ | 0.0709 |
| Liquid–gas momentum flux ratio $q = \rho_j u_j^2/\rho_g u_g^2$ | 2–400 |
| Liquid Weber number $We_j = \rho_j u_j^2 d_j/\sigma$ | 20–1000 |
| Gas Weber number $We_g = \rho_g u_g^2 d_j/\sigma$ | 0–20 |

## 2.2. Measurement Methods

The measurement program includes using high-speed photography for flow visualization and PDA to visualize the droplet characteristics. A high-speed camera (Integrated Design Tools Inc. (IDT), frame rate 6000 fps) is employed to study the disintegration phenomena of a liquid jet by instantaneous photographs. One is captured along the y-axis direction, and the other is captured along the z-axis negative direction, as is shown in Figure 4. The liquid jet breakup images are taken with a Nikon AF-S Micro NIKKOR 60 mm F2.8G Extra-low Dispersion (ED) macro lens, corresponding to 1280 × 1024 pixels. Combined with the MATLAB program, the corresponding image processing method is also used to process the collected images in order to obtain clearer jet breakup images.

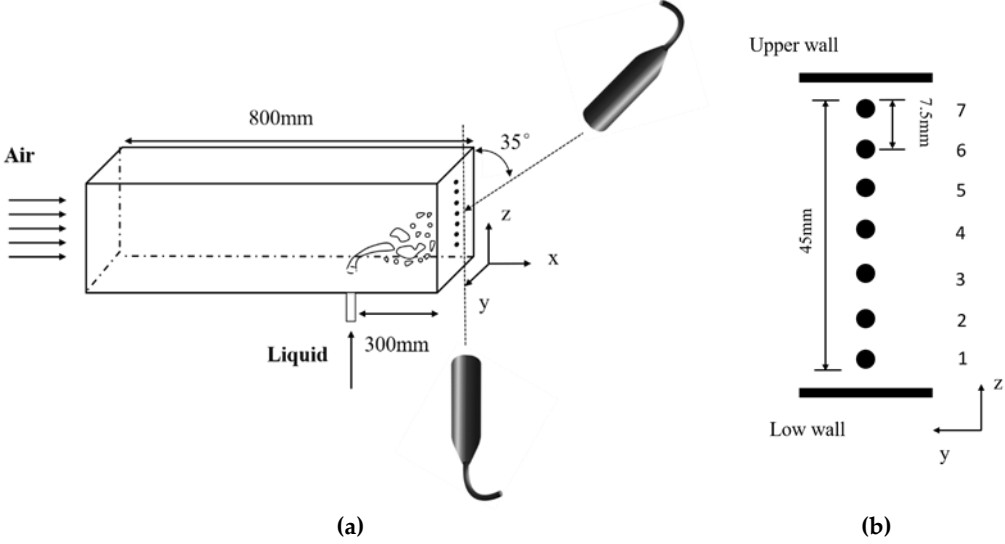

**(a)** **(b)**

**Figure 4.** Schematic of test section, (**a**) is PDA test system and (**b**) is distribution of measuring points.

Droplet sizes and droplet velocities are measured simultaneously by a phase Doppler particle analyzer with a 10-Mw He-Ne laser (PDA, Dantec Dynamics Inc. Copenhagen). Settings of the PDA system are given in Table 2. A beam separator separates the beam into its constituent green (514.5 nm), blue (488.0 nm) and violet (476.5 nm) components. Only the green and the blue beams are used. Data collection on the outlet of the test tube is based on a two-component system from PDA At a distance from the liquid inlet, the diameter and two-dimensional velocity of droplets passing through the

measuring point are measured. The velocity component of the droplet in the direction of the air flow is u, and v is the velocity component of the droplet in the direction of the liquid jet [27].

**Table 2.** Settings and parameters of the PDA system.

|  | Channel 1 | Channel 2 |
|---|---|---|
| **Beam System** | | |
| Wavelength (nm) | 514.5 | 488 |
| Focal length (mm) | 500 | 500 |
| Beam spacing (mm) | 2.2 | 2.2 |
| Beam expander radio | 1.98 | 1.98 |
| Expended beam spacing (mm) | 39.12 | 39.12 |
| Frequency shift (HZ) | 40 | 40 |
| **PDA Receiver** | | |
| Receiver type | Fiber PDA | |
| Scattering angle (deg) | 35 | |
| Receiver focal length (mm) | 500 | |
| Receiver expander radio | 1 | |
| Aperture mask | Mask C | |
| **Particle Properties** | | |
| Particle name | Water | |
| Particle refractive index | 1.334 | |
| Particle specific gravity | 1.0 | |
| Particle kinematic viscosity ($m^2$/s) | 0.001 | |

As shown in Figure 4, in the center line of the test tube outlet, 7 measuring points are uniformly set up along the y–axis direction to acquire the size and velocity characteristics of atomized droplets. Averaged quantities are calculated by collecting 2000 valid sample data for each measurement point.

## 3. Results and Discussion

### 3.1. Breakup Modes

In the present study, the nomenclature suggested by Vich and Ledoux [12] is adopted. Figure 5 shows the characteristics of the bump breakup mode in two directions. From the y-perspective, a stable liquid jet column can be observed near the nozzle when the liquid is injected into the air crossflow. The surface of the jet column has no visible surface fluctuations on the windward side and the leeward side, but the liquid jet column is bent along the airflow direction, as shown in Figure 5a. In the region away from the jet orifice, observations show that some new bumps are generated in the central region of liquid jet direction, and the resulting bumps came off [28], as shown in the red circle in Figure 5b,c. The bump experienced a period of time from birth to shedding, and repeatedly appeared in the next period of time. Under this condition, the effect of the air crossflow on the jet liquid behavior is weak. The time scale of the liquid jet from column to droplets is about 7 ms, and the droplets with a large diameter are produced after the primary breakup. In this breakup mode, the generation and fall-off of such a bump is the most important feature, so the breakup form of the liquid jet column is called bump breakup [26].

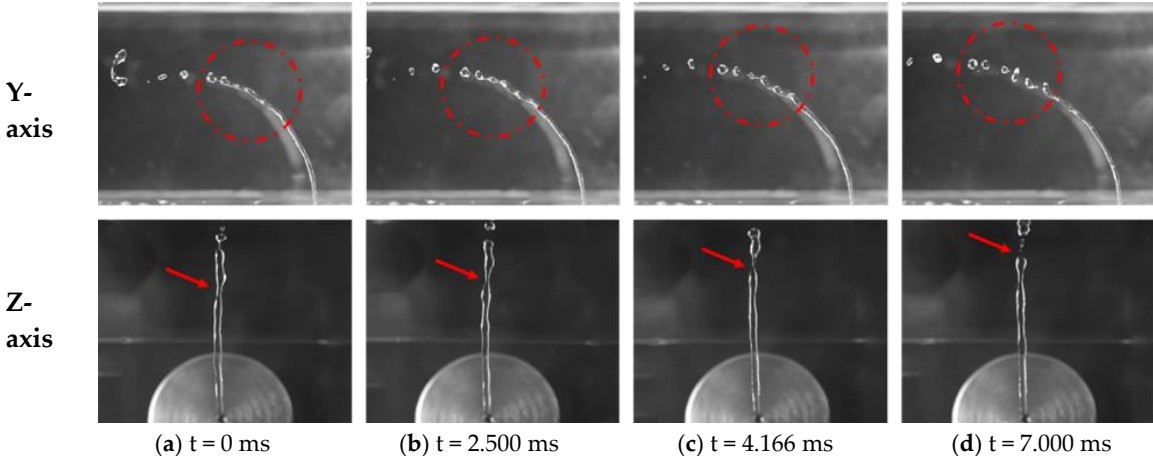

(**a**) t = 0 ms　　　　　(**b**) t = 2.500 ms　　　　　(**c**) t = 4.166 ms　　　　　(**d**) t = 7.000 ms

**Figure 5.** Bump breakup mechanism for $We_g = 1.43$ $We_j = 162.11$ $d_j = 1$ mm.

Figure 6 shows the arcade breakup mode. The overall instability degree of the liquid jet column is more serious than that of the bump breakup, with the increase in $We_g$ in the region away from the jet orifice, as shown in Figure 6b. As a result, many visible fluctuations in the windward side and leeward side are observed when the liquid jet column is further exposed to the airflow [12]. On the z-perspective in Figure 6, the liquid jet column is compressed and divided into dual angled liquid jet columns. With the development of the surface fluctuation of the liquid jet column, the liquid jet column loses its ability to penetrate into the direction of the liquid jet, but forms multiple folds. Multiple liquid column folds are connected by thin liquid lines and elongated along the liquid flow direction (as indicated by the arrow in Figure 6c) until they cannot be maintained. Subsequently, they tear to form tiny droplets. The entire liquid jet breakup process in the air crossflow undergoes a time cycle from jet column to droplets. In one time cycle, the liquid column is bent and flattened into a thin sheet, and then the liquid sheet is torn in the middle to form two arched liquid columns, which further fracture into liquid ligaments and droplets in the air crossflow. It can be observed by high-speed cameras that many arched folds are torn by the air crossflow, resulting in a larger number of small droplets. Since the overall structure of the jet column exhibits arcade-shaped fluctuations, this breakup state can be known as arcade breakup.

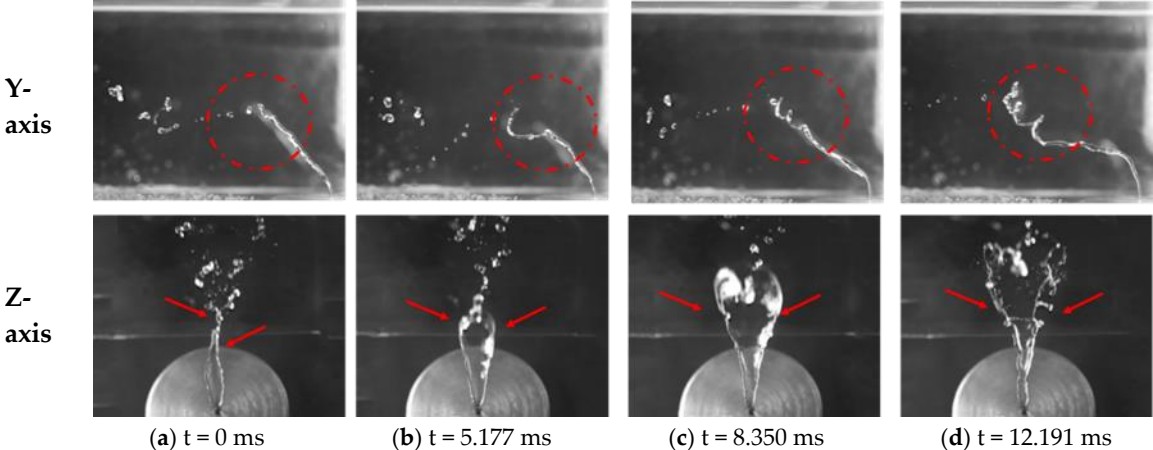

(**a**) t = 0 ms　　　　　(**b**) t = 5.177 ms　　　　　(**c**) t = 8.350 ms　　　　　(**d**) t = 12.191 ms

**Figure 6.** Arcade breakup mechanism for $We_g = 2.90$ $We_j = 245.76$ $d_j = 1$ mm.

Figure 7 displays instantaneous photographs of the bag breakup mode. From the y-perspective, when the liquid jet is injected into the airflow, there is a stable liquid jet column near the liquid jet orifice, which is consistent with the above two breakup modes, except that the distance of the stable liquid column is shorter. The liquid jet column then bends and deforms in the direction of the air

flow. From the z-perspective in Figure 7, the liquid jet column is rapidly compressed and splits into two angled liquid jet columns that have a similar degree of breakup. Furthermore, the instability of the windward side and leeward side of the jet is significant; it makes the jet column flatten under the synergy of the inner and outer surfaces, and sequentially the U-shaped bag ring on the jet column can hardly maintain and ultimately collapses [16]. A large number of broken droplets are produced in a relatively short time cycle, which is exactly expected. In addition, accompanied by the generation of the pocket ring, satellite droplets with much smaller diameters than the jet orifice dimensions are also produced on the leeward surface of the jet column, which is caused by the bursting of the bag ring at the moment of falling off the jet column. Therefore, the generation and shedding of the bag ring are the most important features in this mode [29].

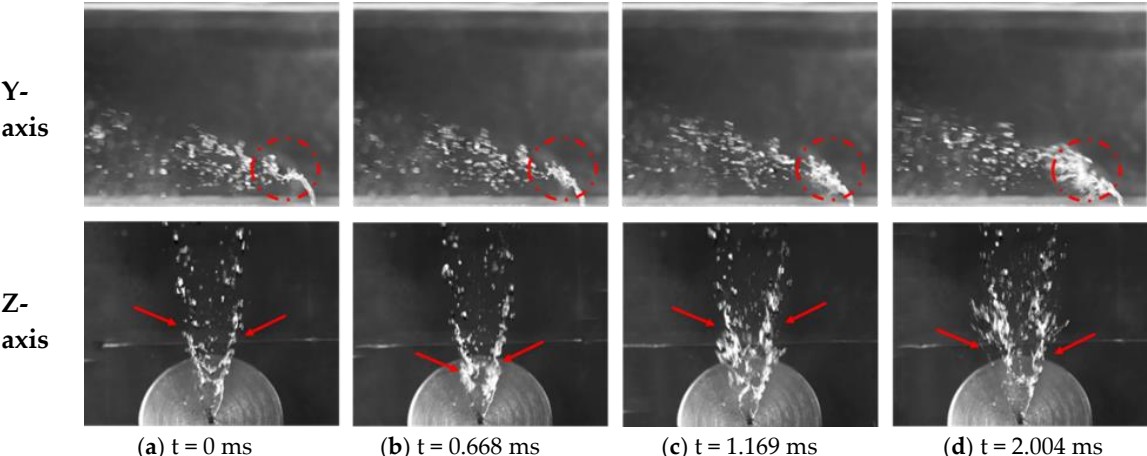

(**a**) t = 0 ms　　　(**b**) t = 0.668 ms　　　(**c**) t = 1.169 ms　　　(**d**) t = 2.004 ms

**Figure 7.** Bag breakup mechanism for $We_g$ = 12.47 $We_j$ = 334.99 $d_j$ = 1 mm.

By analyzing the breakup images of the liquid jet under different gas–liquid flow rates, three liquid jet breakup modes are observed in a low-speed crossflow. The mode and characteristics of the primary breakup of the liquid jet are analyzed through the breakup images of the liquid jet column. Based on a large number of experiments, the influence of gas–liquid conditions on the breakup of liquid jet is summarized, which will be significant for the selection of gas–liquid conditions of tubular gas–liquid contact absorbers in future research.

### 3.2. Liquid Jet Primary Breakup Regimes

Images of the primary breakup of liquid jet for different $We_g$ and $We_j$ are displayed in Figure 8. As long as the liquid jet column is injected into the air crossflow, the air crossflow interacts immediately with the liquid jets. This interaction is responsible for the primary breakup due to undergoing physical processes such as bending, deformation and fracture. The breakup region of the fan-shaped jet flow is formed along the airflow [30]. By comparing images of the same row in Figure 8, it can be found that the primary breakup mode of the liquid jet gradually changes from bump breakup to arcade breakup and finally to bag breakup with an increase in $We_g$ under the condition of the same $We_j$. With the increase in $We_g$ number, the shear force acting on the surface of the liquid jet column by the air crossflow increases gradually, which promotes the bending and breakup process of the liquid column. Therefore, the penetration depth of the jet column along the jet direction decreases with the increase in $We_g$. By comparing the images of liquid jet breakup in the same column in Figure 8, the influence of dimensionless parameter $We_j$ on the liquid breakup process with a constant $We_g$ can be seen. When $We_g$ is 1.8 and $We_j$ changes from 98 to 334, the breakup process of liquid jet is in the bump breakup mode. However, with the increase in $We_j$ value, the penetration depth of the liquid jet in transverse flow increases gradually. When $We_j$ is greater than 253, the liquid jet will be ejected up to the upper wall of the experimental tube section. When $We_j$ is changed from 98 to 334, the liquid jet breakup with

$We_g$ = 2.9 runs the arcade breakup mode. When $We_g$ = 3.9, the breakup mode of liquid jet changes from arcade breakup to bag breakup with an increase in $We_j$. For $We_g$ = 7.1, the breakup process of the liquid jet is performed in the bag breakup mode with various values of $We_j$.

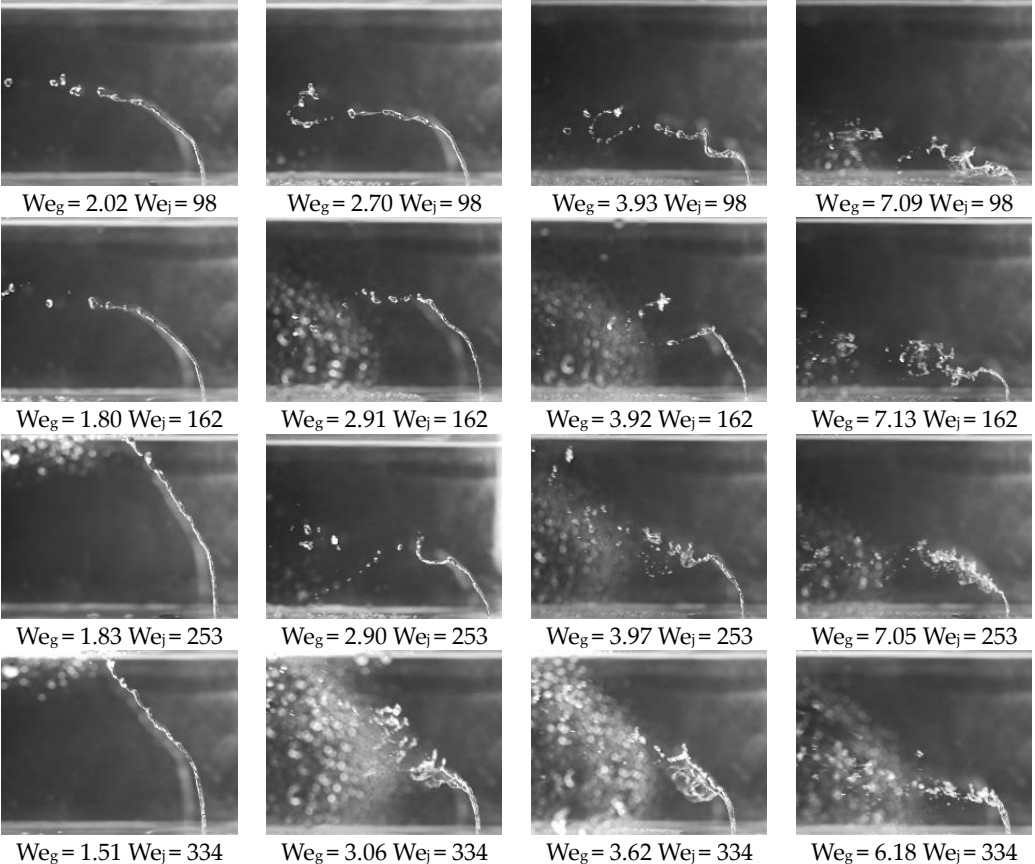

| $We_g$ = 2.02 $We_j$ = 98 | $We_g$ = 2.70 $We_j$ = 98 | $We_g$ = 3.93 $We_j$ = 98 | $We_g$ = 7.09 $We_j$ = 98 |
| $We_g$ = 1.80 $We_j$ = 162 | $We_g$ = 2.91 $We_j$ = 162 | $We_g$ = 3.92 $We_j$ = 162 | $We_g$ = 7.13 $We_j$ = 162 |
| $We_g$ = 1.83 $We_j$ = 253 | $We_g$ = 2.90 $We_j$ = 253 | $We_g$ = 3.97 $We_j$ = 253 | $We_g$ = 7.05 $We_j$ = 253 |
| $We_g$ = 1.51 $We_j$ = 334 | $We_g$ = 3.06 $We_j$ = 334 | $We_g$ = 3.62 $We_j$ = 334 | $We_g$ = 6.18 $We_j$ = 334 |

**Figure 8.** Images of the primary breakup process of the liquid jet under different $We_g$ and $We_j$.

In order to better predict the mode characteristics of jet primary fractures under different working conditions, the flow visualization experiments are performed over a wide range of some parameters to identify the regions of parameter ranges dominated by different breakup states. In conjunction with the above typical characteristics of liquid jet breakup, there are two modes of liquid jet breakup under the action of a low-speed air crossflow: column breakup and bag breakup. The experimental results show that the column breakup could be divided into two distinct characteristics of bump breakup and arcade breakup under different gas–liquid working conditions. With $We_g$ as the abscissa and $We_j$ as the ordinate, a breakup regime map of a liquid jet in a low-speed air crossflow is attained [26], as shown in Figure 9. It can be clearly observed from the Figure 9 that there is an obvious transition boundary between column breakup and bag breakup, and $We_g = 10^{\frac{3.5-\log We_j}{1.93}}$. Under the condition of higher $We_j$ and $We_g$, bag breakup plays a dominant role, whereas under the condition of lower $We_j$ and $We_g$, column breakup is dominant. Additionally, when $We_g > 2.8$, the liquid jet breakup state exhibits characteristics of the arcade breakup mode. When $We_g < 2.8$, it exhibits the characteristics of the bump breakup mode. For a certain $We_j$, the liquid jet breakup has multiple breakup modes as the abscissa $We_g$ changes. However, for a certain $We_g$, there is a single determined breakup mode for the liquid jet breakup as the ordinate $We_j$ changes. Hence, compared with the different values of $We_j$, the liquid jet breakup mode is more sensitive to the variations of $We_g$. The reason is the increase in the transverse flow pressure promotes the growth of the surface disturbance waves, and make the liquid jet column windward and leeward more unstable. Thus, $We_g$ can be considered as the most effective

dimensionless parameter to determine the breakup mode for the liquid jets injected into a low-speed air crossflow.

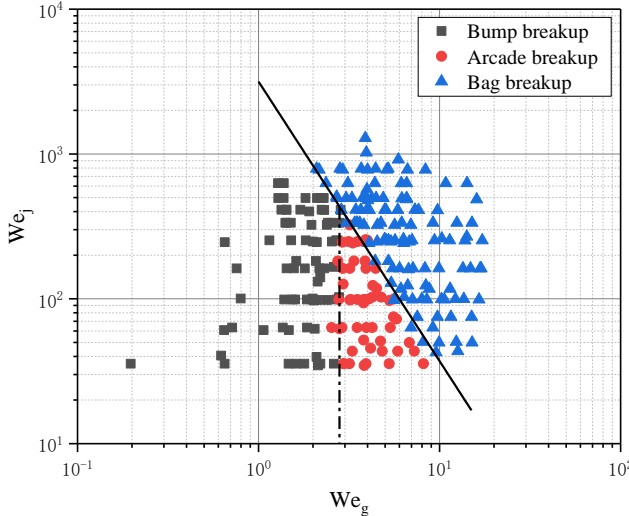

**Figure 9.** Primary breakup regime map of a liquid jet in a low–speed air crossflow based on $We_g$ and $We_j$.

### 3.3. Droplet Sizes Characteristics

Droplet sizes and velocities are measured simultaneously by PDA. Droplet diameter is one of the most important parameters to evaluate the atomization performance of a liquid jet. Sauter mean diameter (SMD) is commonly used to describe the size characteristics of atomized droplets [31].

Figure 10 depicts the SMD droplet size distribution along the z direction for different $We_j$ with the same $We_g$. The z direction is the liquid jet direction, and z = 0 represents the first measurement point at 2.5 mm from the jet orifice. It can be seen from Figure 10 that the droplet size distribution curve along the z direction presents an approximate oblique-I-type for different $We_j$. Specifically, the droplet size is smaller near the jet orifice and larger away from the jet orifice. The reason for this result is that the unstable surface fluctuation caused by the difference in the horizontal flow pressure occurs on the leeward and windward side of the jet column, when the liquid jet is injected into the transverse airflow. This unstable surface fluctuation causes the deformation of the jet column's leeward side inward depression. The cross-section of the liquid jet column is deformed, which further enables tiny droplets to form on the leeward surface. As can be seen in Section 3.2, with a constant $We_g$, the penetration depth and exposure time of the liquid jet in the transverse flow increase with the increase in $We_g$. The length of the leeward side of liquid jet column that produces tiny droplets becomes larger afterwards.

However, part of the liquid jet column will be sprayed on the upper wall to form a liquid film when the penetration depth of the liquid jet further increases. The liquid film is attached to the upper wall along the air flow direction and will not be atomized into small droplets. Then, the efficiency of liquid jet atomization will be decreased. With a constant $We_g$, the droplet size distribution curve gradually shifts to the decreasing direction of droplet size as $We_j$ increases. With the increase in $We_j$, the upper atomized droplets are mainly formed by the liquid column spraying onto the upper wall of the experimental tube, and the atomized droplets at the lower measuring point are mainly shed from the leeward side of the liquid column. When the PDA system is applied, the measurement time of each measurement point with 2000 particle size samples is significantly increased. As a result, most of the liquid in the jet column is ejected to the upper wall surface without directly participating in the gas–liquid interaction.

As $We_g$ increases, the difference of droplet size between upper and lower test points decreases gradually, and the droplet size distribution tends to be uniform. The reason is that the liquid jet penetration decreases with the increase in $We_g$, and the breakup area of liquid column gradually moves to the center of the experimental tube section.

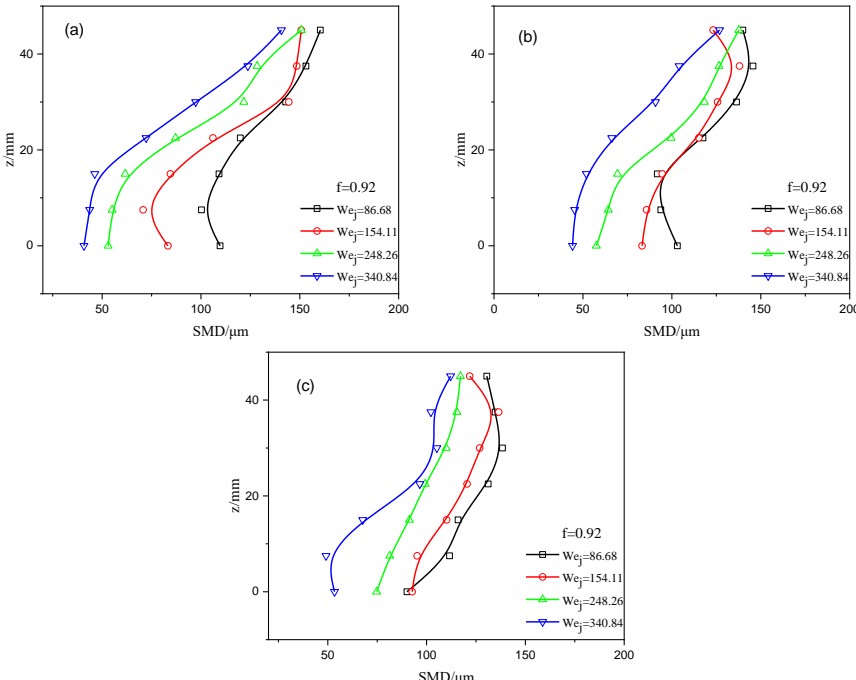

**Figure 10.** Droplet size distribution along the z direction for different $We_j$ at the same $We_g$, (**a**) $We_g$ = 4.64, (**b**) $We_g$ = 6.19, (**c**) $We_g$ = 7.55.

With the further increase in $We_g$, the droplet size distribution curve will change from oblique-I-type to C-type. The C-type droplet size distribution is better visible in Figure 11. As shown in Figure 11, the droplet size in the center of the tube is the largest, and the droplet size near the upper and lower wall of the pipe is smaller. The distribution of droplet size measured in the experimental tube shows a trend of being larger in the middle and smaller on both sides. When $We_g > 9.51$, the breakup mode of the liquid jet changes into the bag breakup mode. The liquid jet column bends and breaks quickly in the transverse airflow. The breakup process of the liquid jet mainly occurs in the center of the experimental tube section. Therefore, the amount of liquid injected up to the upper wall decreases significantly, and the efficiency of liquid jet atomization increases. The central position of the experimental tube is the primary breakup zone of the liquid jet, and the complex liquid column breakup process produces a large number of small droplets and large droplets [26]. As a result, the droplet size measured in the center of the experimental tube section is larger. Small droplets detected near the lower wall of the experimental tube are produced by the breakup of the leeward side of the liquid column near the jet hole and the turbulent diffusion of small droplets with the air flow. Small droplets detected near the upper wall of the experimental tube occur partly from the external breakup of the jet plume and partly from the turbulent diffusion of small droplets with the air flow. When $We_g > 9.51$, $We_j$ has little influence on droplet size distribution, and the droplet size is in the range of 50–150 μm.

From the above analysis, it can be concluded that there is a critical point of $We_{gc}$ = 7.55 for the droplet distribution characteristics. When $We_g < We_{gc}$, the droplet size distribution curve along the z direction is oblique-I-type. When $We_g > We_{gc}$, the distribution curve of droplet size along z direction is C-type. In order to ensure that the tubular gas–liquid atomization mixed contactor has better atomization performance and atomization efficiency, the $We_g$ of the liquid jet breakup zone should be more than 7.55.

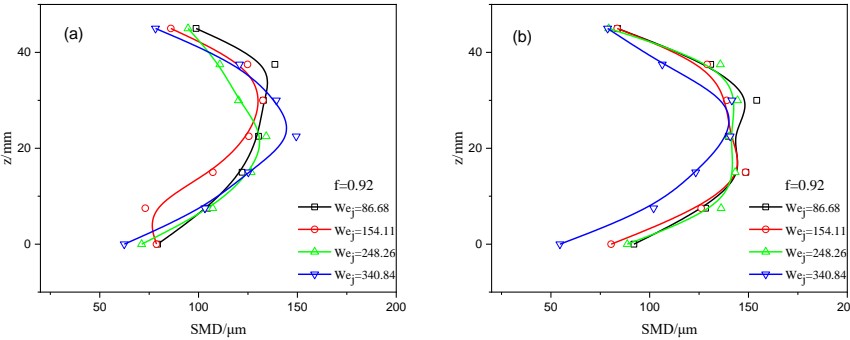

**Figure 11.** Droplet size distribution along the z direction for different $We_j$ at the same $We_g$, (**a**) $We_g = 9.51$, (**b**) $We_g = 12.19$.

Furthermore, in order to determine the distribution characteristics of droplet size collected at different measuring points in the finite sample volume, Figures 12 and 13 shows the statistical distribution of droplet size at different measuring points. When the horizontal coordinates are logarithmic, the distribution of the number of droplets at the measuring point is in the form of lognormal distribution. The distribution of droplet number on the measuring point can be described by Equation (1) [18].

$$f(x; \mu; \sigma) = \frac{1}{x\sigma\sqrt{2\pi}} e^{\frac{(\ln x - \mu)^2}{2\sigma^2}} \tag{1}$$

where $f(x; \mu; \sigma)$ is probability density function; x is the sample particle diameter; $\mu$ is the geometric mean diameter; $\sigma$ is the geometric standard deviation.

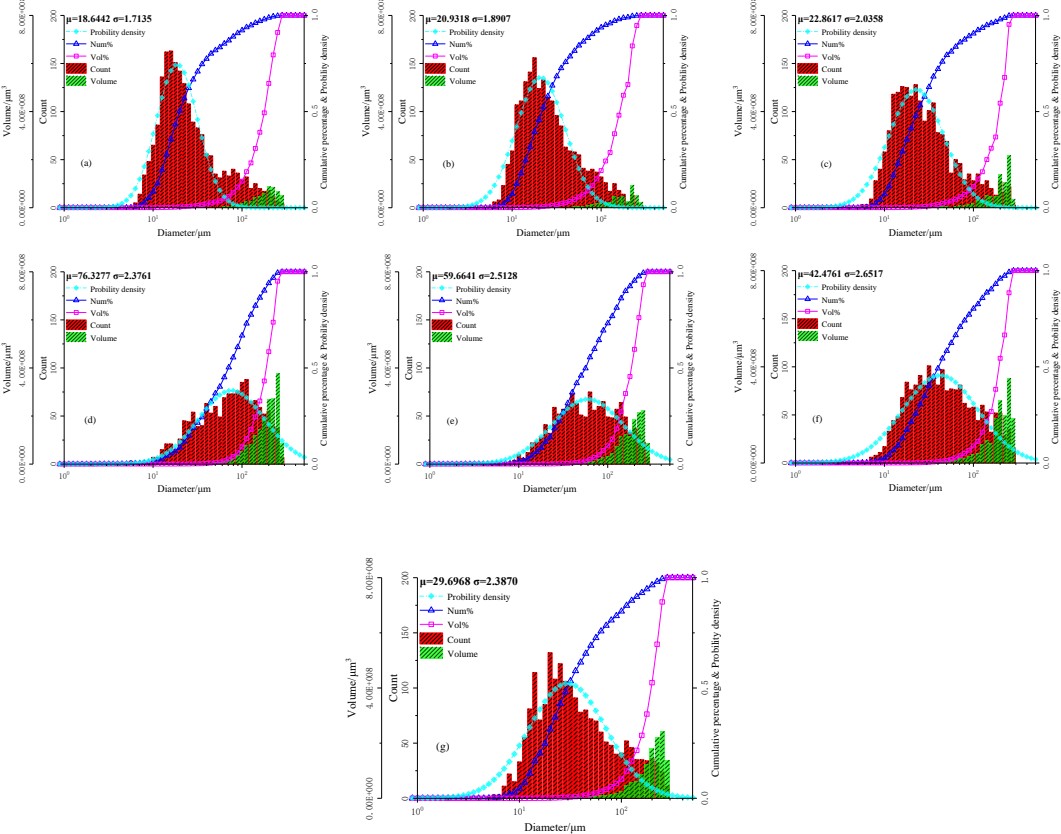

**Figure 12.** Statistical characteristics of droplet size at different measuring points along the z direction for the $We_g = 9.51$, $We_j = 154.11$, (**a**) z = 0, (**b**) z = 7.5, (**c**) z = 15, (**d**) z = 22.5, (**e**) z = 30, (**f**) z = 37.5, (**g**) z = 45.

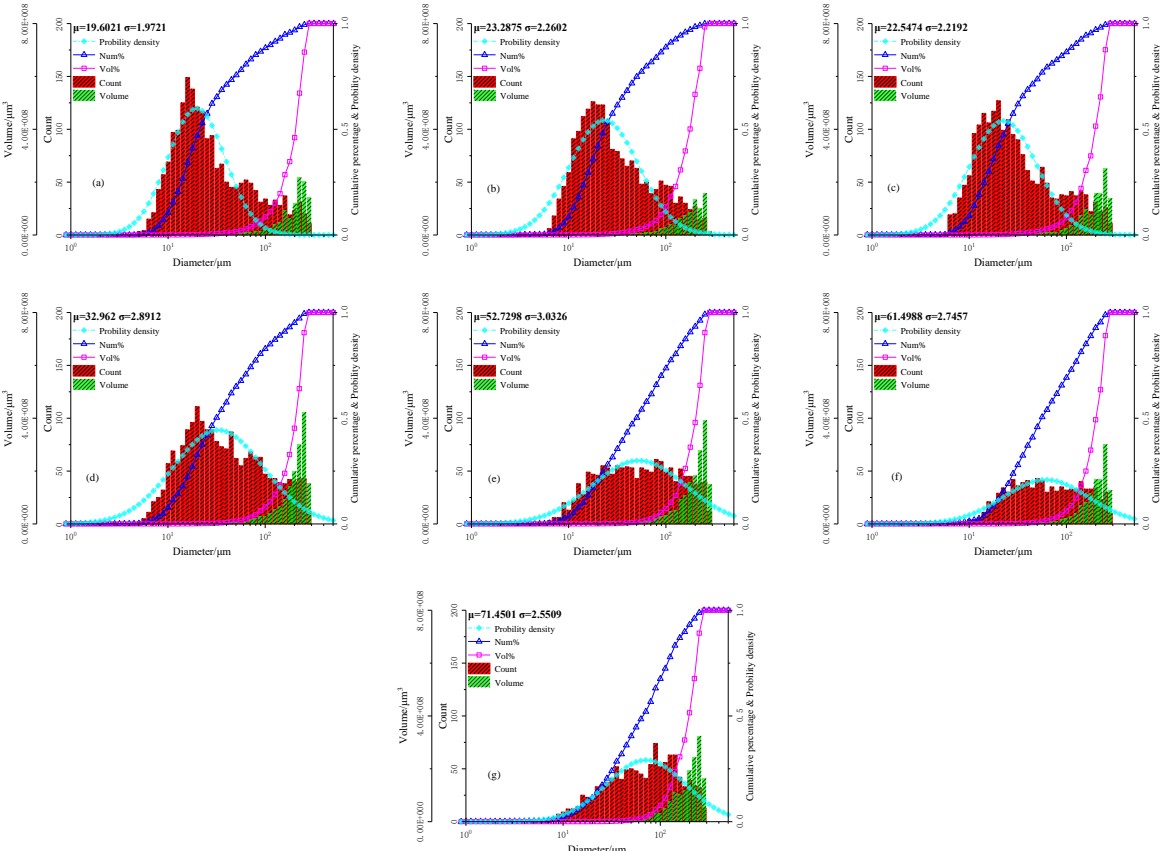

**Figure 13.** Statistical characteristics of droplet size at different measuring points along the z direction for the $We_g = 6.19$, $We_j = 248.25$, (**a**) z = 0, (**b**) z = 7.5, (**c**) z = 15, (**d**) z = 22.5, (**e**) z = 30, (**f**) z = 37.5, (**g**) z = 45.

Figure 12 shows the probability density function curve of droplet number distribution at different measuring points along the liquid jet direction (z direction). The geometric mean value of liquid droplets increases first and then decreases, and it indicates that the droplets near the upper and lower walls are generally of a smaller particle size. The geometric standard deviation firstly decreases and then increases. A reduction in the geometric standard deviation indicates that the droplets are more concentrated in a smaller size range. Therefore, the droplet size distribution along the liquid jet direction is C-type, as mentioned above. At the position near the upper and lower walls, the histogram is "thinner" and "higher". Near the middle area, the histogram is "thicker" and "shorter".

Figure 13 shows the statistical characteristics of the oblique-I-type distribution at different measuring point for $We_g = 6.19$, $We_j = 248.25$. As can be seen from the trends of the droplet number distribution histogram and the droplet volume distribution histogram, the overall histogram gradually moves to the right, and the geometric average diameter of the droplet and the volume of the large particle size gradually increase with the increase in the distance in the liquid jet direction (the z value increases). The shape of droplet number distribution histogram changed from tall and thin to short and thick, respectively.

### 3.4. Droplet Velocity Characteristics

By employing PDA to measure the droplet velocity at several test points, the distribution of droplet velocity at different positions in the spray field can be obtained. Figure 14a shows the dependence of the average velocity along the liquid jet direction for a constant $We_g$ and a different $We_j$. When $We_g$ remains constant, the droplet velocity distribution for different $We_j$ looks similar. This suggests that the atomized droplets accelerate gradually in the air flow, and the droplet velocity is basically same as

the gas velocity at the outlet of experimental tube section. Although the liquid jet flow rate is different, the droplet velocity in the far field of the spray is basically similar under the same gas flow condition. Figure 14b shows the droplet velocity along the liquid jet direction for a constant $We_j$ and a different $We_g$. The magnitude of droplet velocity in the far field of the spray increases as the $We_g$ increases from 4.64 to 12.19. $We_g$ has a decisive effect on the droplet velocity distribution in the outlet section of test tube. The effect of liquid column on the flow diminishes with the increase in the liquid jet streamwise distance, x, due to the column breakup. This, therefore, leads to a more direct momentum exchange between droplets and the cross airflow, and thus the acceleration of droplets.

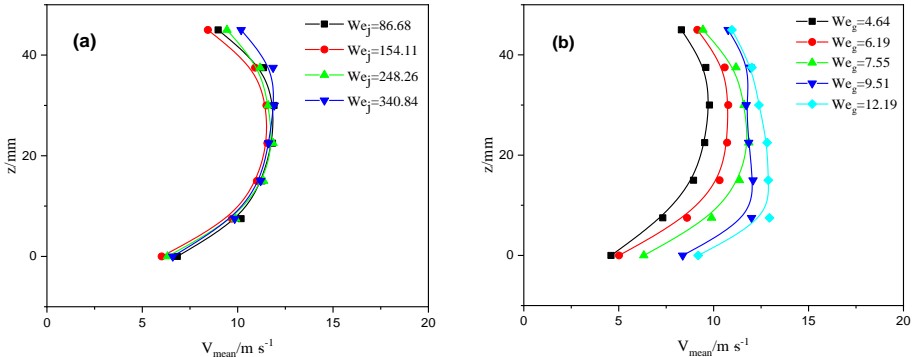

**Figure 14.** Droplet velocity distribution along the z direction: (**a**) $We_g = 7.55$, (**b**) $We_j = 154.1$.

## 4. Conclusions

The breakup process and atomization characteristics of liquid jets injected into a low-speed air crossflow in the finite tube space are experimentally investigated. Firstly, high-speed photographic observation is carried out to study the deformation and breakup properties of a liquid jet with different values of $We_g$ and $We_j$. Secondly, detailed droplet sizes and droplet velocities are obtained for various liquid jets using PDA.

The unsteady surface fluctuation on the windward and leeward sides is the main reason for the bending, deformation and fracture of the liquid jet. The primary breakup mods of the liquid jet injected into a low-speed air crossflow mainly consists of column breakup and bag breakup. Column breakup could be further divided to two distinct characteristics of bump breakup and arcade breakup under different gas–liquid working conditions. The $We_g$–$We_j$ regime map of primary breakup processes of the liquid jet in a low–speed air crossflow is obtained based on the visual observation. The column/bag breakup borderline is given by $We_g = 10^{\frac{3.5-\log We_j}{1.93}}$ $We_g = 2.8$ is the borderline between the bump and arcade breakup modes.

The droplet distribution characteristic has a critical point of $We_{gc}$: $We_{gc} = 7.55$. When $We_g < We_{gc}$, the droplet size distribution curve along the z direction is oblique-I-type. For $We_g > We_{gc}$, the droplet size distribution curve is C-type. When $We_g > 7.55$, the breakup mode of the liquid jet changes into the bag breakup mode. The liquid jet column bends and breaks quickly in the transverse airflow. The breakup process of the liquid jet mainly occurs in the center of the experimental tube section. $We_j$ has little effect on droplet size distribution, and the droplet size is in the range of 50–150 μm. In order to ensure that the tubular gas–liquid contactor has better atomization performance and atomization efficiency, the $We_g$ of the liquid jet breakup zone should be more than 7.55.

Droplet velocities increase significantly with an increase in the $We_g$ of the crossflow, but the change in droplet velocity when increasing the $We_j$ is not obvious. $We_g$ has a decisive effect on the droplet velocity distribution in the outlet section of the test tube.

**Author Contributions:** L.K.: conceptualization, data curation, writing—review and editing. T.L.: investigation, data curation, writing—original draft. J.C.: resources, funding acquisition. K.W.: supervision. H.S.: methodology. All authors have read and agreed to the published version of the manuscript.

**Funding:** This research was funded by the National Natural Science Foundation of China (21808015).

**Conflicts of Interest:** The authors declare no conflict of interest.

## Nomenclature

| | |
|---|---|
| $A_g$ | cross-sectional area of the square tube |
| $A_j$ | cross-sectional area of the liquid jet orifice |
| $d_j$ | liquid jet orifice diameter |
| Oh | Ohnesorge number |
| PDA | phase-Doppler anemometry |
| q | liquid–gas momentum flux ratio |
| $Q_j$ | measured liquid flow rate |
| $Q_g$ | measured air crossflow rate |
| SMD | Sauter mean diameter |
| $u_j$ | liquid jet velocity |
| $u_g$ | air crossflow velocity |
| $We_g$ | gas Weber number |
| $We_j$ | liquid jet Weber number |
| $\rho_j$ | liquid density |
| $\rho_g$ | air density |
| $\sigma$ | surface tension |

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
