# Peer review of "Breakup Processes and Droplet Characteristics of Liquid Jets Injected into Low-Speed Air Crossflow"

_processes, doi:10.3390/pr8060676_

Round 1

Reviewer 1 Report

This paper describes the droplet breakup processes and characteristics of droplets when fluid injected into low speed air flow. The authors also use parameters such as Weg to describe results. The paper is novel in its design and written very well. It is suitable for publication in Processes after few suggestions listed below.

Abstract: Define Weg

Figure numbering is wrong and misplaced such as Fig.3 and 6 are missing. 

Figure 14 and 15 can be combined.

The paper should be limited to 8 figures as there are currently too many figures and this limits the readers interest. Other should be included into the Supplementary file.

Author Response

Dear editors and reviewers:

Thanks you for your constructive comments and suggestions concerning our manuscript entitled “Breakup processes and droplet characteristics of liquid jet injected into low-speed air crossflow” (Manuscript: processes-814128). Those comments are all valuable and helpful for revising and improving our paper. According to your suggestions, we have modified our manuscript carefully. In this revised version, changes in our manuscript were all highlighted within the document by using the "Track Changes" function in Microsoft Word. Point-by-point responses to the reviewers are listed below this letter. please refer to the email in the attachment.

Thank you and sincere regards.

Sincerely,

Lan

Reviewer 2 Report

The paper presents the results of experimental research on the decay of a liquid stream as a result of its introduction into a flowing gas. Based on the results of the experiment, a map of primary breakdown processes was created, and more importantly the boundaries between these areas were determined. From the experimental point of view, all studies were carried out in accordance with all the principles. Also, the interpretation of the results obtained does not raise any objections. In addition, the presented results have significant application potential.

I am not a linguist, but some English-language formulations seemed to me contrary to the prevailing principles. So maybe it would be worthwhile for the specialist to review the text of the paper.

Author Response

Dear editors and reviewers:

Thank you for your constructive comments and suggestions concerning our manuscript entitled” Breakup processes and droplet characteristics of liquid jet injected into low-speed air crossflow” (Manuscript No.814128). Those comments are all valuable and helpful for revising and improving our paper. According to your suggestions, we have modified our manuscript carefully. Furthermore, we have invited a native English speaking colleague to help correct grammatical and expressive errors in our manuscript. In this revised version, changes to our manuscript were all highlighted within the document by using the “Track Changes” function in Microsoft Word. Point-by-point responses to the reviewers are listed below this letter.Please refer to the email in the attachment.

Thank you and sincere regards.

Sincerely,

Lan
